# Al Foil-Supported Carbon Nanosheets as Self-Supporting Electrodes for High Areal Capacitance Supercapacitors

**DOI:** 10.3390/molecules28041831

**Published:** 2023-02-15

**Authors:** Jiaojiao Zheng, Bing Yan, Li Feng, Qian Zhang, Jingquan Han, Chunmei Zhang, Weisen Yang, Shaohua Jiang, Shuijian He

**Affiliations:** 1International Innovation Center for Forest Chemicals and Materials, Co-Innovation Center of Efficient Processing and Utilization of Forest Resources, College of Materials Science and Engineering, Nanjing Forestry University, Nanjing 210037, China; 2College of Science, Nanjing Forestry University, Nanjing 210037, China; 3Institute of Materials Science and Devices, School of Materials Science and Engineering, Suzhou University of Science and Technology, Suzhou 215009, China; 4Fujian Key Laboratory of Eco-Industrial Green Technology, College of Ecology and Resources Engineering, Wuyi University, Wuyishan 354300, China

**Keywords:** self-supporting electrode, carbon nanosheets, hydrothermal reaction, chemical vapor deposition, areal capacitance, supercapacitors

## Abstract

Self-supporting electrode materials with the advantages of a simple operation process and the avoidance of the use any binders are promising candidates for supercapacitors. In this work, carbon-based self-supporting electrode materials with nanosheets grown on Al foil were prepared by combining hydrothermal reaction and the one-step chemical vapor deposition method. The effect of the concentration of the reaction solution on the structures as well as the electrochemical performance of the prepared samples were studied. With the increase in concentration, the nanosheets of the samples became dense and compact. The CNS-120 obtained from a 120 mmol zinc nitrate aqueous solution exhibited excellent electrochemical performance. The CNS-120 displayed the highest areal capacitance of 6.82 mF cm^−2^ at the current density of 0.01 mA cm^−2^. Moreover, the CNS-120 exhibited outstanding rate performance with an areal capacitance of 3.07 mF cm^−2^ at 2 mA cm^−2^ and good cyclic stability with a capacitance retention of 96.35% after 5000 cycles. Besides, the CNS-120 possessed an energy density of 5.9 μWh cm^−2^ at a power density of 25 μW cm^−2^ and still achieved 0.3 μWh cm^−2^ at 4204 μW cm^−2^. This work provides simple methods to prepared carbon-based self-supporting materials with low-cost Al foil and demonstrates their potential for realistic application of supercapacitors.

## 1. Introduction

With the sustaining consumption of fossil fuels, the energy shortage crisis has been considered as an issue that cannot be neglected. To relieve this problem, new energy conversion and storage technologies have been developed, such as supercapacitors and batteries. In comparison with traditional capacitors and batteries, supercapacitors possess higher power density, longer cycle life, higher charging/discharging efficiency, higher and safety performance, as well as produce less environmental pollution [1,2,3,4]. However, the large-scale practical application of supercapacitors is still hindered by the high capital cost of active materials and the relatively low energy density [5,6]. To overcome such challenges, much effort has been undertaken to develop novel electrode materials with special nanostructures and high performance [7,8,9].

Self-supporting electrode materials are promising candidates with novel nanostructures and an active substance for high performance supercapacitors [10,11]. Recently, self-supporting electrode materials have attracted a lot of attention because they do not need extra current collectors and any additional binders as well as conductive additives, which promotes the utilization efficiency of not only the active substance, but also the electron transmission rate [9,12,13]. There are two main methods to prepare self-supporting electrode materials: bottom up and top down. The former method usually assembles the nanostructured materials, such as graphene nanosheets, carbon quantum dots, carbon nanotubes, and so on, into free-standing structures by filtering, wet spinning, and hydrothermal reactions, as well as electrostatic spinning, while the latter method takes the three-dimensional substrates as the base to grow the secondary nanostructures on their surface by hydrothermal/solvothermal reactions, chemical vapor deposition (CVD), atomic layer deposition (ALD), and so on [14,15]. 

Compared to the bottom-up methods, the top-down methods are more convenient. At present, the top-down methods mainly grow active materials on non-metallic substrates with self-supporting structures (carbon cloth [16], carbon paper [17], carbon foam [7], carbon sponge [18], and so on) and metal substrates as electrode materials. The metal substrates possess the merits of excellent electrical conductivity, outstanding mechanical strength, superior flexibility, high stability, and cost-effectiveness, thus many self-supporting electrode materials based on metal substrates have been designed and optimized in the past decade [19].

In the reported literature works, the metal substrates commonly used for the preparation of self-supporting electrode materials are gold (Au) [20], silver (Ag) [21], copper (Cu) [22], nickel (Ni) [23], iron (Fe) [24], and aluminum (Al) [25,26], among others. An ideal metal substrate used to prepare a self-supporting electrode is supposed to have the advantages of high electrical conductivity, low cost, and a simple synthesis processes. The substrates of Au, Ag, Cu, and Al possess high direct current electrical conductivity values of 4.10 × 107, 6.30 × 107, 5.96 × 107, and 3.50 × 107 S m^−1^, respectively, which are higher than those of the substrates of Ni, Pt, and Pd [19]. Considering the cost of the substrates, the noble metal substrates are very low-cost in nature. Although Cu substrates exhibit high electrical conductivity, the prices of Cu substrates are relatively high and the content of Cu in the earth is relatively low. Al is not only the richest metal in the crust, but also possesses a much lower cost compared with Cu and other substrates. Moreover, the density of Al is much lower than that of other metal substrates, which is beneficial to decrease the total mass of the energy storage devices. In recent years, many Al-based self-supporting electrode materials have been prepared by hydrothermal reactions [15], CVD [25,27], wet chemical methods [28], and so on, and possess high electrochemical performance. Therefore, Al substrates are promising candidates to prepare self-supporting electrode materials for the application of supercapacitors.

In this work, novel self-supporting electrode materials with carbon nanosheets deposited on Al foil were prepared by combining hydrothermal reaction and the CVD method. The highlight of this work is the construction of hierarchical nanostructures on the Al foil to increase the load of active material. To begin with, zinc compound nanosheets were grown on the Al foil substrate by the hydrothermal method. Subsequently, carbon was coated on the as-prepared Al foil substrate by the CVD method with acetonitrile serving as the carbon source. When applied to the coin-type symmetric supercapacitors, the optimized sample of CNS-120 exhibits an areal capacitance of 6.82 mF cm^−2^ at a current density of 0.01 mA cm^−2^. It also displayed ultrahigh rate performance with an areal capacitance of 3.07 mF cm^−2^ at 2 mA cm^−2^ and high energy density of 5.9 μWh cm^−2^ at a power density of 25 μW cm^−2^. Moreover, the coin-type supercapacitors of CNS-120 can power light-emitting diodes and digital watches, demonstrating the promising application of supercapacitors in practice.

## 2. Results and Discussion

### 2.1. Morphology and Characterization of Samples

#### 2.1.1. Diagram of the Preparation of Electrode Materials

The synthesis process of the self-supporting Al/ZnO/C electrode combining hydrothermal reaction and the chemical vapor deposition (CVD) method is illustrated in Figure 1. The zinc compound nanosheets were grown on the Al foil substrate directly via a facile hydrothermal method. Subsequently, the carbon was coated on the as-prepared Al foil substrate by the CVD method with acetonitrile serving as the carbon source (about 0.21 mL min^−1^). As shown in Appendix A, the surface of Al foil became off-white after hydrothermal treatment, which turned to black when the CVD process finished. It proved that carbon was coated on the Al foil successfully. What is more, the CNS-120 electrode exhibited excellent mechanical capacity and superior flexibility in Appendix A.

The masses of CNS-30, CNS-60, and CNS-120 were measured in six samples. The average values of their mass per unit area were 8.5, 8.2, and 7.8 mg cm^−2^, respectively. In order to estimate the active material content of CNS-30, CNS-60, and CNS-120, their thermal decomposition behaviors were investigated the thermogravimetric analyses (TGA). As shown in Appendix A, the weight loss of the samples before 50 °C was caused by the loss of the inherent free water. With the increase in temperature (50 to 100 °C), the slight increase in mass is due to the fresh Al being oxidized to Al_2_O_3_ when exposed to the air atmosphere. The weight loss of the samples from 100 to 500 °C was attributed to the carbon materials being oxidized to CO_2_. The residual amounts of CNS-30, CNS-60, and CNS-120 were 96.20%, 95.90%, and 96.90%, respectively. The peak values of CNS-30, CNS-60, and CNS-120 were 100.22%, 100.16%, and 100.50%, respectively. Therefore, the carbon yields of CNS-30, CNS-60, and CNS-120 were 4.02%, 4.26%, and 3.60%, respectively. The active materials of the single electrode used in the coin-type symmetric supercapacitors of CNS-30, CNS-60, and CNS-120 were 0.34, 0.35, and 0.28 mg cm^−2^, respectively.

#### 2.1.2. SEM

The SEM images in Figure 2 display the morphology of all samples after hydrothermal and CVD treatment. The zinc compound precursors exhibit the ordered thin nanosheets grown on the surface of Al foil substrate in Figure 2a,c,e. With the increase in concentration, the nanosheets of NS-30, NS-60, and NS-120 became increasingly compact. After CVD and the carbonization process, as shown in Figure 2b,d,f, the nanosheets of the composites became more compact and thinner as a result of the action of mechanical stress and strain. Significantly, CNS-120 is composed of composite nanosheets and cracks, which is because the expansion rate of composite nanosheets is different from that of the Al foil substrate. This result is in accordance with the digital photo of CNS-120 shown in Appendix A, whose size is a little smaller than that of NS-120. The cracks on the surface of CNS-120 may be beneficial to the electrolyte storage and ion transport.

#### 2.1.3. XRD, Raman, and TEM

To further study the composition of the nanosheets grown on the surface of the Al foil substrate, XRD, Raman spectroscopy, and TEM were performed, and the results are elucidated in Figure 3. Figure 3a exhibits the XRD patterns of Al foil, NS-120, and CNS-120. It is obvious that the Al foil without any treatment possesses four peaks ((111), (200), (220), and (311)), with the 2θ located at 38.3°, 44.5°, 64.9°, and 78.1°, respectively, which is consistent with the standard Al diffraction lines (JCPDS data: PDF#85-1327) [15,29]. All four peaks are also observed in NS-120 and CNS-120 because of the Al foil substrate. As displayed in the XRD patterns, NS-120 possesses diffraction peaks of Zn-Al-LDH (Zn_6_Al_2_(OH)_16_CO_3_·4H_2_O) located at 9.6°, 19.5°, 34.0°, and 59.9°, which are the diffraction at (003), (006), (009), and (110), respectively, in accordance with the LDH database (JCPDS data: PDF# 38-0486) [30]. Besides, there also are four peaks of NS-120 located at 31.5°, 34.2°, 36.1°, and 56.4°. According to the JCPDS data (PDF#80-0074), they are the diffraction peaks at (100), (002), (101), and (110), respectively, of ZnO [31,32]. Therefore, after the hydrothermal reaction, the nanosheets on the surface of CNS-120 are composed of ZnO and Zn-Al-LDH. The mechanisms of reactions during the hydrothermal treatment are listed as follows [15,33,34]:(CH_2_)_6_N_4_ + 6H_2_O → 6HCHO + 4NH_3_,(1)
NH_3_ + H_2_O → NH_4_^+^ + OH^−^,(2)
HCHO + 2OH^−^ → CO_3_^2−^ + 2H_2_,(3)
Zn^2+^ + 4NH_3_ → [Zn(NH_3_)_4_]^2+^,(4)
[Zn(NH_3_)_4_]^2+^ + 4OH^−^ → [Zn(OH)_4_]^2−^ + 4NH_3_,(5)
Zn^2+^ + 4OH^−^ → [Zn(OH)_4_]^2−^,(6)
[Zn(OH)_4_]^2−^ → ZnO + H_2_O + 2OH^−^,(7)
2Al + 2OH^−^ + 6H_2_O → 2[Al(OH)_4_]^−^ + 3H_2_,(8)
2[Al(OH)_4_]^−^ + 6[Zn(OH)_4_]^2−^ + CO_3_^2−^ + 4H_2_O → Zn_6_Al_2_(OH)_16_CO_3_·4H_2_O + 16OH^−^,(9)

Apparently, after the CVD process, there is a new diffraction peak at 24.0° in CNS-120, which is related to the (002) amorphous graphite reflection [35,36,37]. The d_002_ of the carbon on CNS-120 is 0.37 nm, calculated using Bragg’s equation. This demonstrates that the carbon was coated on the Al substrate successfully. What is more, the diffraction peaks of Zn-Al-LDH disappear in the XRD pattern of CNS-120, which is related to the decomposition of Zn-Al-LDH at a high temperature. During the process, LDH was decomposed into ZnO, Al_2_O_3_, and CO_2_, as well as H_2_O, and Al_2_O_3_ was reduced to Al through the carbon and hydrogen decomposed from acetonitrile. The reaction processes are shown as follows [15,38]:2CH_3_CN → 4C + N_2_ (g) + 3H_2_ (g),(10)
Zn_6_Al_2_(OH)_16_CO_3_·4H_2_O → Zn_6_Al_2_(OH)_16_CO_3_ + 4H_2_O (g),(11)
Zn_6_Al_2_(OH)_16_CO_3_ → 6ZnO + Al_2_O_3_ + CO_2_ (g) + 8H_2_O (g),(12)
2C + Al_2_O_3_ → 2Al + CO_2_ (g) + CO (g),(13)
3H_2_ + Al_2_O_3_ → 2Al + 3H_2_O (g),(14)

The Al foil substrate without any treatment has no peaks in the spectrum. After the hydrothermal process, NS-120 possesses two peaks located at 547 cm^−1^ and 484 cm^−1^, which correspond to the stretching vibration of Zn-O-Al and Al-O-Al, respectively [39]. According to the previous studies, the peak located at 713 cm^−1^ accounts for the absorption of CO_2_ when NS-120 was exposed to air [30,40]. The peak located at 1051 cm^−1^ may be caused by the stretching vibration of nitrate [30,39]. As displayed in Figure 3b, the peaks of Zn-Al-LDH disappeared in the spectrum of CNS-120 when the CVD process finished. Instead, there are two new peaks located at 1345 and 1594 cm^−1^ in the Raman spectrum of CNS-120, which are the D-band (sp^3^-hybridized, reflecting the defects and disorder of carbon material) and G-band peaks (the graphitic sp^2^-hybridized) [41,42]. The relative intensity ratio of the D band and G band (I_D_/I_G_) can reflect the defect and disorder level of the carbon materials [41,43,44]. The I_D_/I_G_ value of CNS-120 is 1.21, which demonstrates that the carbon deposited on CNS-120 is amorphous. It also confirms that the carbon was deposited on the surface of the Al foil substrate successfully. The big peak observed at 332 cm^−1^ was related to the transverse optical mode (A_1_(TO)) of ZnO [45]. The whole analysis of Raman spectra is in line with the XRD results.

The structure of CNS-120 was further investigated using TEM at different magnifications (Figure 3c,d). CNS-120 possesses two obvious lattice fringes with distances of 0.25 and 0.28 nm in the high-resolution TEM image (Figure 3d), which correspond to the (101) and (100) plane of ZnO, respectively [32,46]. Moreover, there is a phase with a spacing of 0.37 nm in CNS-120. It is considered as the amorphous carbon, which has relatively ordered microcrystallites [47]. The TEM results confirm the XRD and Raman results as well. Considering the above analysis, the possible microstructure of the electrode is shown in Appendix A. The surface area of the electrode can be enhanced by the coaxial Al/ZnO core/shell nanosheets, providing more space for loading carbon materials. After the deposition of the carbon on the surface of nanosheets, more active sites can be provided for reversible redox reactions, which benefits the enhancement of the electrochemical performance of the electrode.

#### 2.1.4. XPS

The elemental composition on the surface of CNS-120 was analyzed by XPS (Figure 4). The Zn, Al, C, N, and O elements can be detected in the full spectrum of XPS (Figure 4a), where the contents of C, N, and O are 3.7, 0.21, and 2.56 at.%, respectively (Appendix A). Two peaks at a binding energy (BE) of 1021.9 and 1044.9 eV are considered as Zn 2p_3/2_ and Zn 2p_1/2_, respectively, indicating that Zn^2+^ presents in CNS-120 [32,48]. There are also three peaks located at about 10.9, 89.5, and 140.0 eV, corresponding to Zn 3d, Zn 3p, and Zn 3s, respectively. Furthermore, there are two peaks originating from Al 2p and Al 2s at 74.3 and 119.3 eV, respectively, indicating the presence of Al^3+^ in CNS-120 [39]. Apparently, CNS-120 possesses three peaks located at about 285, 400, and 531 eV, which correspond to C 1s, N 1s, and O 1s, respectively [49]. The contents of C 1s, N 1s, and O 1s in CNS-120 are 3.7 at.%, 0.21 at.%, and 2.56 at.%, respectively (Appendix A). The presence of N and O atoms may contribute to the enhancement of the capacitance of electrode materials because of the Faraday reaction [49]. As presented in Figure 4b, the high-resolution C 1s spectrum is fitted into three peaks at 284.8, 286.2, and 289.4 eV, which correspond to C=C (73.4%), C-O (19.0%), and O=C-O (7.6%) bonds, respectively [50]. N 1s can be divided into three peaks (Figure 4c), corresponding to pyridinic nitrogen (N-6, 398.4 eV, 29.2%), pyrrolic nitrogen (N-5, 400.3 eV, 58.9%), and pyridine-N-oxide (N-O, 403.9 eV, 11.9%) [51,52]. O 1s of CNS-120 can be assigned to three peaks at 530.8, 532.2, and 533.4 eV, which represent Zn-O (59.9%), C=O (33.5%), and C-O (6.6%), respectively [36,46].

### 2.2. Electrochemical Performance of Coin-Type Supercapacitors

To investigate the electrochemical performance of the as-prepared electrode materials, symmetric coin-type supercapacitors were assembled with 1 M Et_4_NBF_4_/PC as the organic electrolyte as well as NKK-TF4030 as the membrane, and the test results are depicted in Figure 5 and Appendix A. Figure 5a presents the CV curves of CNS-30, CNS-60, and CNS-120 at the scan rate of 50 mV s^−1^, which exhibit quasi-rectangular shapes, indicating the double-layer capacitive behavior [53]. Meanwhile, the CV curve of CNS-120 possesses a larger integrated area than that of CNS-30 and CNS-60, indicating the largest areal capacitance of CNS-120 [54]. As displayed in Appendix A, with the increasing scan rate (10~300 mV s^−1^), the CV curves of CNS-30 turn into the shuttle shape, which is related to the enhancive ion transport resistance. At the scan rate of 10~300 mV s^−1^, CNS-60 and CNS-120 maintain a quasi-rectangular shape (Appendix A)). The rapid diffusion of electrolytes and charge transfer are demonstrated [55]. When the scan rate increased from 400 mV s^−1^ to 1 V s^−1^, the CV curves of CNS-120 gradually became the shuttle shape, demonstrating the best rate performance among the three electrodes (Figure 5b). The areal capacitances (C_S_) of CNS-30, CNS-60, and CNS-120 were calculated from the CV results at different scan rates. The C_S_ of CNS-30, CNS-60, and CNS-120 was 2.42, 2.83, and 3.24 mF cm^−2^, respectively, at the scan rate of 10 mV s^−1^ (Appendix A). When the scan rate was increased to 300 mV s^−1^, the C_S_ of CNS-30, CNS-60, and CNS-120 was 0.29, 1.58, and 1.85 mF cm^−2^, respectively.

As displayed in Figure 5c, the GCD curves of CNS-30, CNS-60, and CNS-120 exhibit a nearly symmetric triangular shape at a current density of 0.01 mA cm^−2^, which suggests the outstanding capacitive performance [54]. What is more, the GCD curve of CNS-120 shows a longer discharging time than that of CNS-30 and CNS-60, which confirms that CNS-120 has a larger areal specific capacitance. When the current density reached 0.1 mA cm^−2^, the GCD curves of CNS-60 and CNS-120 maintained the triangular symmetric shape with a slight IR drop (Appendix A). However, the charge/discharge segment of CNS-30 deviates from a straight line and displays a large IR drop. These demonstrate that CNS-120 possesses better capacitive performance than CNS-30 and CNS-60. At current densities from 0.01 mA cm^−2^ to 1 mA cm^−2^, the GCD curves of CNS-120 and CNS-60 maintain a nearly symmetric triangular shape, while that of CNS-30 possesses a distorted triangular shape with a large IR drop (Appendix A). The results suggest that CNS-60 and CNS-120 possess better electrochemical and rate performance than CNS-30.

The areal capacitances (C_A_) of CNS-30, CNS-60, and CNS-120 were also calculated based on the GCD results at different current densities. As shown in Figure 5e, the C_A_ of CNS-120 was 6.82 mF cm^−2^ at a current density of 0.01 mA cm^−2^ and decreased to 3.07 mF cm^−2^ at 2 mA cm^−2^, while the C_A_ of CNS-60 was 6.04 mF cm^−2^ at 0.01 mA cm^−2^ and decreased to 2.37 mF cm^−2^ at 2 mA cm^−2^. Besides, the C_A_ of CNS-30 was 5.36 mF cm^−2^ at 0.01 mA cm^−2^. When the current density was increased to 1 mA cm^−2^, the C_A_ of CNS-30 was decreased to nearly 0 mF cm^−2^. The CNS-120 exhibits a higher areal capacitance than CNS-30 and CNS-60, which is consistent with the CV test results. The areal capacitance obtained from GCD is higher than that obtained from CV, attributed to a longer charge/discharge time for GCD, which ensures more time for generating pseudocapacitance. The maximum areal capacitance of CNS-120 is superior to some previously reported electrode materials, such as MPC (6.3 mF cm^−2^ at 0.8 mA cm^−2^) [61], HCSs (6.1 mF cm^−2^ at 0.5 mA cm^−2^) [62], 3D FC-CNT@P (5.53 mF cm^−2^ at 0.1 mA cm^−2^), and N-doped sucrose carbon (3.9 mF cm^−2^ at 5 mV s^−1^) [63], as well as laser-induced graphene (4 mF cm^−2^ at 0.01 mA cm^−2^) [58]. More details of the comparisons between CNS-120 and the previously reported material electrodes are summarized in Appendix A [64,65]. What is more, CNS-120 also showed the best rate performance among the electrode materials. With the increasing concentration of the hydrothermal solution, the electrode materials possess better electrochemical performance. This is due to the increasing number of nanosheets on the electrode, which can provide more active sites. During the charging and discharging process, the charge storage mechanism of the electrode is the coexistence of the electric double layer capacitance and pseudocapacitance. As shown in the rate performance results, the capacitance decreases with the increases in current density. On the one hand, owing to the increase in current density, the charge/discharge rate was too fast and the redox reaction of N and O functional groups in active materials could only partially occur, leading to the reduction in the contribution of pseudocapacitance. On the other hand, the acceleration of the charge/discharge rate also led to the decrease in the micropore utilization rate of the porous carbon layer on the electrode surface, resulting in the reduction in the contribution of the electric double layer capacitance.

The symmetric coin-type supercapacitor of CNS-120 delivers the maximum energy density of 5.9 μWh cm^−2^ at a power density of 25 μW cm^−2^ and still achieves 0.3 μWh cm^−2^ at a high power density of 4204 μW cm^−2^ (Figure 5f). Such superb electrochemical performance exceeds that of some previously reported electrode materials, such as 3D FC-CNT@P (0.49 μWh cm^−2^ at 40 μW cm^−2^) [55], laser-induced graphene (4.5 μWh cm^−2^ at 905 μW cm^−2^; 0.256 μWh cm^−2^ at 110 μW cm^−2^) [58,60], and G-CNT-5 (1.36 μWh cm^−2^ at 26 μW cm^−2^) [59].

The Nyquist plots of CNS-30, CNS-60, and CNS-120 are shown in Figure 6, which is beneficial to understanding the capacitive behaviors of all of the samples. All of the plots contain a semicircle in the high-frequency region and a line perpendicular to the real axis in the low-frequency region, which reflect the charge transfer resistance (R_ct_, the diameter of semicircle) and the diffusion resistance (R_d_) [25,66,67], respectively. The equivalent series resistance (ESR), the real axis intercept of the semicircle in the high-frequency region, reflects the inner resistance of materials, the interface contact resistance of the electrode and electrolyte, as well as the electrolyte ionic resistance [68,69]. The ESR values of CNS-30, CNS-60, and CNS-120 are 1.04, 1.17, and 1.16 Ω, respectively (Appendix A). In addition, the R_ct_ values of CNS-30, CNS-60, and CNS-120 are 1533, 15.24, and 149.3 Ω, respectively. The trend of the R_ct_ values is to first decrease and then increase with the increase in concentration, which is related to the surface structures of the samples. As demonstrated in the SEM images, with the increase in concentration, the nanosheets of NS-30, NS-60, and NS-120 became increasingly compact (Figure 2a,c,e). This means that, under the same conditions, the more nanosheets that grew on the surface of the Al foil, the higher the carbon content coated on the surface, which is helpful for the enhancement of conductivity. Therefore, the R_ct_ values of CNS-60 and CNS-120 are much lower than that of CNS-30. However, the nanosheets on the surface of CNS-120 are too close (Figure 2f), which may make it difficult for electrolyte ions to enter the gaps. Thus, the R_ct_ value of CNS-120 is higher than that of CNS-60. As shown in Figure 6a, in comparison with CNS-30 and CNS-60, CNS-120 displayed a more vertical line at a low frequency. This indicates that CNS-120 exhibits more outstanding capacitive behavior and higher areal capacitance, which is in line with the CV and GCD results.

To characterize the electrode cycling behaviors, the durability of CNS-30, CNS-60, and CNS-120 was investigated by GCD tests at 0.1 mA cm^−2^ (Figure 7). After 5000 cycles, the areal capacitances of CNS-30, CNS-60, and CNS-120 remained at 86.79%, 96.95%, and 96.35% of initial capacitance, respectively. The results demonstrate that CNS-120 possesses outstanding stability.

As displayed in Figure 8, the devices composed of three coin-type supercapacitors of CNS-120 can light up an ‘NJFU’ logo consisting of 35 parallel light-emitting diodes (LEDs) after charging at 2.5 V. What is more, the three parallel devices can power the digital watch for 40 min after charging at 2 V (Figure 8b,c). Thus, the coin-type supercapacitors of CNS-120 are promising candidates to apply for energy conversion and storage in practice.

## 3. Materials and Methods

### 3.1. Materials

Al foil (Al-1235, thickness: 20 μm) was obtained from Foshan Zhongji ximi New Material Co., Ltd. Zinc nitrate (Zn(NO_3_)_2_·6H_2_O) and hexamethylenetetramine (HMT) were purchased from Nanjing Chemical Reagent Co., Ltd. Acetonitrile was available from Macklin Co., Ltd. The organic electrolyte with 1 M Et_4_NBF_4_ in PC (propylene carbonate) and the cellulose membrane (NKK-TF4030) were supplied by Canrd Co., Ltd. The deionized water was homemade. All agents were of analytical grade, without any purification.

### 3.2. Preparation of Zinc Compound Nanosheets

Typically, zinc compound nanosheets were prepared by the hydrothermal method. Herein, 30 mL of HMT solution was added to 30 mL of zinc nitrate aqueous (30, 60, and 120 mmol) drop-by-drop by stirring. The molar ratio of HMT and Zn(NO_3_)_2_·6H_2_O was 1:1. Then, the homogeneous solution and Al foil substrates (4.1 cm × 6.6 cm) were transferred into the Teflon-lined vessel, which was sealed in a stainless steel autoclave and kept in an oven at 70 °C for 12 h. After cooling to room temperature, the Al foil substrates were taken out and washed with deionized water. Finally, the treated Al foil substrates were dried at 60 °C overnight. The obtained treated Al foil substrates with zinc compound nanosheets were named NS-x, where x means the concentration of zinc nitrite solution.

### 3.3. Preparation of Al/ZnO/C Nanosheets

To prepare carbon nanosheets on the samples’ surface, Al foil substrates with zinc compound nanosheets were placed at the center of the tube furnace. Before heating, nitrogen (N_2_) was introduced into the tube furnace to remove the air at a speed of 200 mL min^−1^. Then, 15 min later, N_2_ was used as the carrier gas as well. The acetonitrile serving as the carbon source was introduced into the tube furnace at a rate of about 0.21 mL min^−1^ when the temperature reached 600 °C (the heating rate: 5 °C min^−1^). Then, 1 h later, the acetonitrile was turned off while the furnace was kept at 600 °C for 1 h. When the furnace cooled to the room temperature, the samples were taken out and named CNS-30, CNS-60, and CNS-120.

### 3.4. Characterization Methods

All of the samples were characterized at an acceleration voltage of 15 kV with scanning electron microscopy (SEM, Phenom XLG2). The morphology of CNS-120 was observed by transmission electron microscopy (TEM, JEM-2100 UHR). The thermal decomposition behaviors of CNS-30, CNS-60, and CNS-120 were investigated by thermogravimetric analyses (TGA, NETZSCH TG 209F3) in an air atmosphere with a heating rate of 10 °C min^−1^. The X-ray diffraction (XRD, Ultima IV) patterns of CNS-120 were achieved from 10 to 80° at a sweep speed of 10° min^−1^ at 30 mA and 40 kV. Raman spectroscopy (DXR523) was utilized to characterize the sample with a wavelength of 523 nm laser. The chemical compositions of CNS-120 were measured by X-ray photoelectron spectra (XPS, Thermo Fisher Nexsa).

The interlayer spacing of carbon materials (d_002_ (nm)) was obtained from Bragg’s law (Equation (15)) based on the XRD results.
2d_002_ sinθ = λ,(15)
where θ (°) is the diffraction angle and λ (0.15418 nm) is the wavelength of X-ray with copper Kα radiation.

### 3.5. Electrochemical Performance Measurements

The electrochemical performance of as-prepared CNS-x was tested on the CHI 760E electrochemical workstations (Shanghai ChenHua, Shanghai, China) at room temperature. CNS-30, CNS-60, and CNS-120 were assembled into coin-type symmetric two-electrode supercapacitors with 1 M Et_4_NBF_4_/PC serving as the organic electrolyte. The electrode was cut into a wafer with a diameter of 12 mm. Two pieces of electrode were separated by a piece of cellulose membrane (NKK-TF4030, d = 16 mm). Cyclic voltammetry (CV) as well as galvanostatic charge/discharge (GCD) were conducted at the potential window of 0 to 2.5 V and current density of 0.01 to 2 mA cm^−2^ (0.029 to 0.714 mA mg^−1^). Electrochemical impedance spectroscopy (EIS) was obtained at the frequency varying from 10^−2^ to 10^5^. The areal specific capacitance of the electrode was calculated by Equations (16) and (17):C_A_ = 2I × Δt/(S × ΔV),(16)
C_S_ = S_CV_/2SkΔV_CV_(17)
where C_A_ (mF cm^−2^) is the area-specific capacitance of the electrode, I (A) is the discharge current, Δt (s) is the discharge time, S (cm^2^) is the area of a single electrode, ΔV (V) is the potential window excluding the IR drop, C_S_ (mF cm^−2^) is the area specific capacitance of the electrode calculated from the CV results, S_CV_ (AV) is the integral area of the CV curve, k (mV s^−1^) is the scan rate of the CV test, and ΔV_CV_ (V) is the potential window of CV (total voltage range).

The power density and energy density of the assembled supercapacitors were calculated by the following equation:E = C_A_ × ΔV^2^/(2 × 3.6),(18)
P = 3600 × E/Δt,(19)
where E (μWh cm^−2^) and P (μW cm^−2^) are the energy density and power density, respectively, of the assembled coin-type symmetric supercapacitors; C_A_ (mF cm^−2^) is the area-specific capacitance; ΔV (V) is the potential window excluding the IR drop; and Δt (s) is the discharge time.

## 4. Conclusions

In conclusion, self-supporting electrode materials with carbon nanosheets grown on Al foil were prepared successfully by combining hydrothermal reaction and the one-step CVD method. In the process, the nanostructures of the self-supporting electrodes were regulated as well as controlled by adjusting the concentration of the hydrothermal reaction solution and the acetonitrile serving as the carbon source. The influence of the concentration of the hydrothermal reaction solution on the nanostructures and electrochemical performance of all the samples was revealed. It turns out that, with the increase in concentration, the nanosheets of NS-30, NS-60, and NS-120 became increasingly more compact. After CVD and carbonization, when used as coin-type supercapacitors, CNS-120 exhibited excellent electrochemical performance among all of the prepared materials. The areal capacitance of CNS-120 was 6.82 mF cm^−2^ at a current density of 0.01 mA cm^−2^ and still maintained 3.07 mF cm^−2^ at a current density of 2 mA cm^−2^. It also exhibited an energy density of 5.9 μWh cm^−2^ at a power density of 25 μW cm^−2^ and still achieved 0.3 μWh cm^−2^ at a high power density of 4204 μW cm^−2^. Besides, CNS-120 displayed good cyclic stability with a capacitance retention of 96.35% after 5000 cycles at 0.1 mA cm^−2^. What is more, three parallel coin-type supercapacitors of CNS-120 were able to light up 35 LED lamps and power a digital watch for 40 min, which indicated that self-supporting electrode CNS-120 is a promising candidate for realistic application. In general, carbon-based self-supporting electrodes based on Al foil prepared by effective hydrothermal reaction and the one-step CVD method provide a promising candidate for applications in energy conversion and storage.

## Figures and Tables

**Figure 1 molecules-28-01831-f001:**
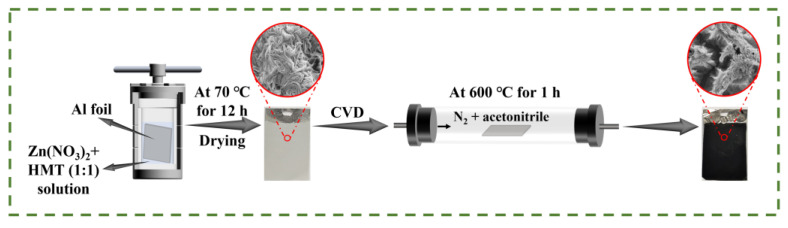
Diagram of the preparation of the self-supporting Al/ZnO/C electrode materials.

**Figure 2 molecules-28-01831-f002:**
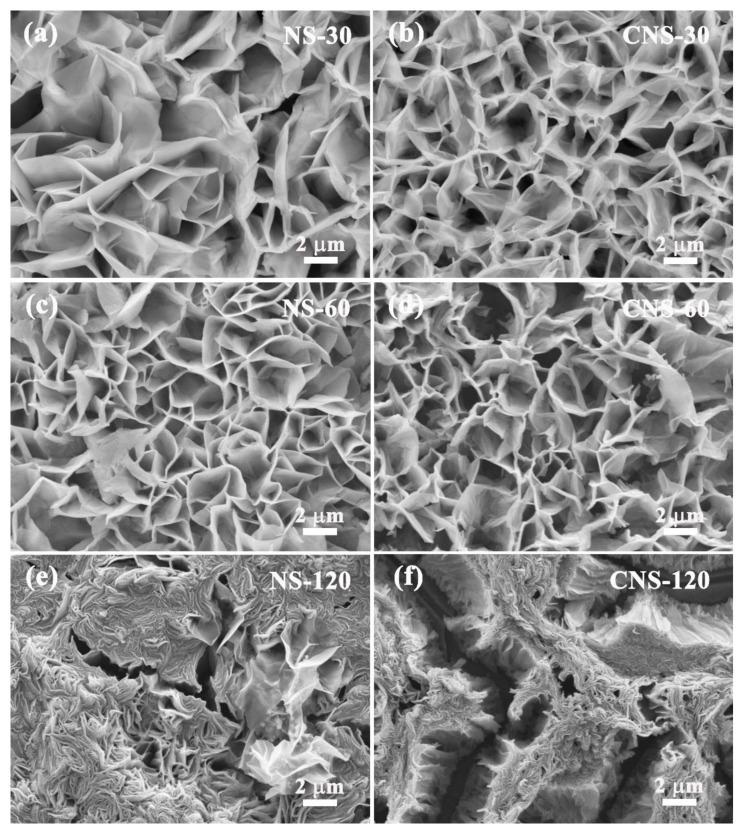
SEM images of all samples. (**a**) NS-30. (**b**) CNS-30. (**c**) NS-60. (**d**) CNS-60. (**e**) NS-120. (**f**) CNS-120.

**Figure 3 molecules-28-01831-f003:**
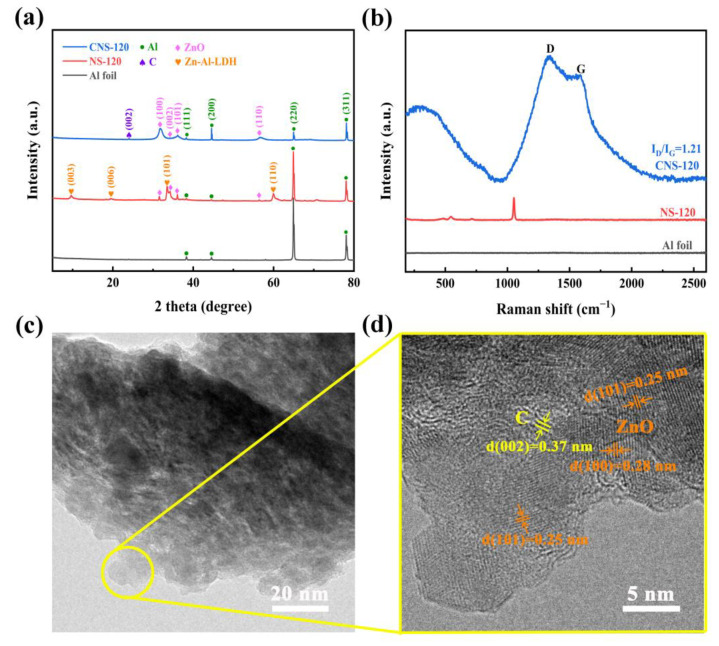
Characterization of NS-120 and CNS-120. (**a**) XRD patterns of Al foil, NS-120, and CNS-120. (**b**) Raman spectra of Al foil, NS-120, and CNS-120. (**c**) and (**d**) TEM images of CNS-120.

**Figure 4 molecules-28-01831-f004:**
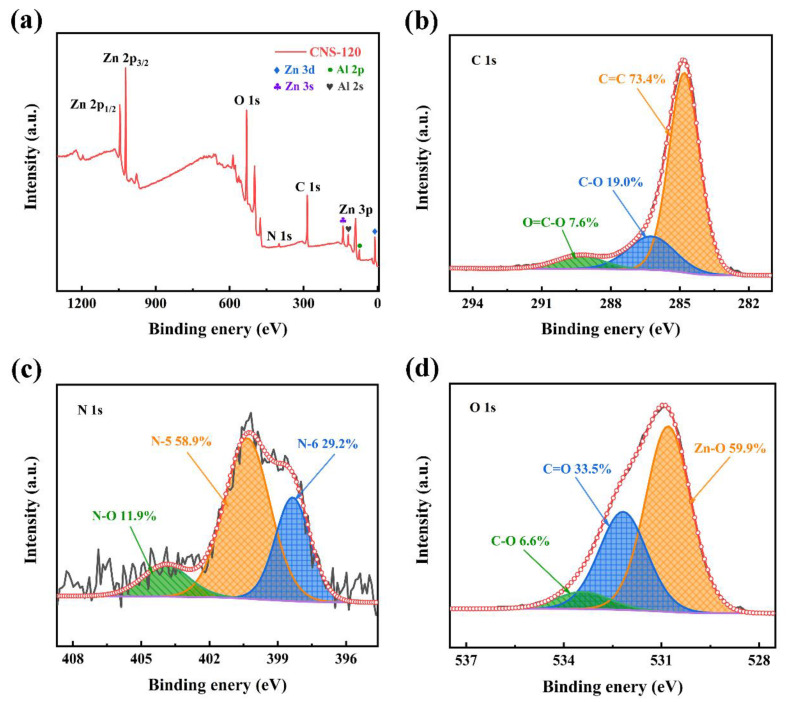
XPS spectra of CNS-120. (**a**) XPS survey spectrum of CNS-120. (**b**) C 1s and (**c**) N 1 and (**d**) O 1s high-resolution XPS spectra of CNS-120.

**Figure 5 molecules-28-01831-f005:**
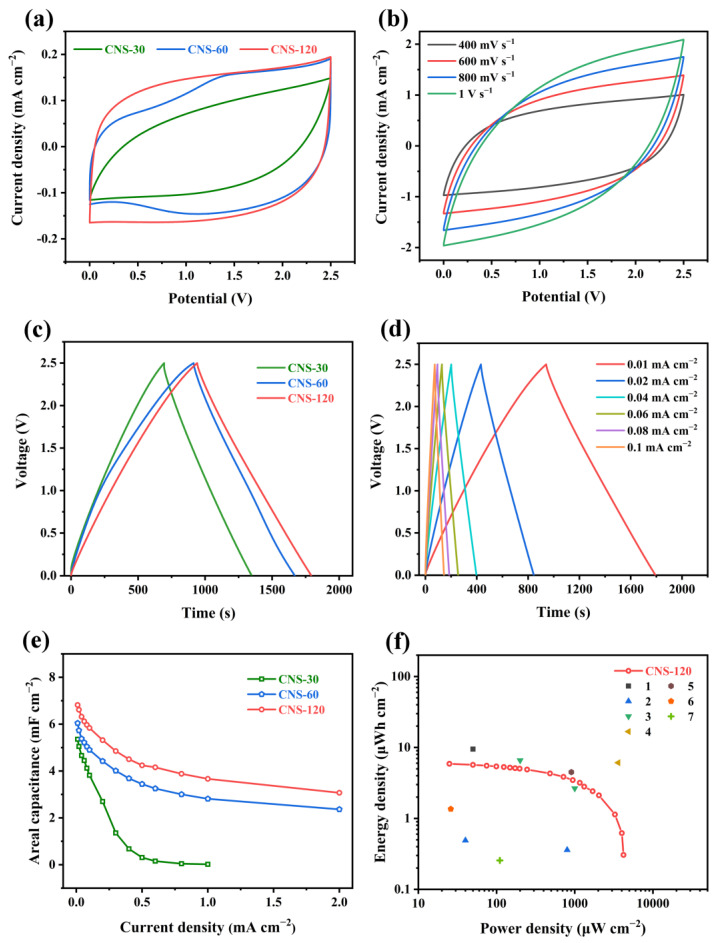
Electrochemical performance of the samples in the two-electrode system. (**a**) CV curves of CNS-30, CNS-60, and CNS-120 at a scan rate of 50 mV s^−1^. (**b**) CV curves of CNS-120 at scan rates from 400 mV s^−1^ to 1 V s^−1^. (**c**) GCD curves of CNS-30, CNS-60, and CNS-120 at a current density of 0.01 mA cm^−2^. (**d**) GCD curves of CNS-120 at current densities from 0.01 to 0.1 mA cm^−2^. (**e**) Areal capacitance of CNS-30, CNS-60, and CNS-120 at different current densities. (**f**) Ragone plot of CNS-120 in comparison with reported materials: (1) PC-IA-MSC [54], (2) 3D FC-CNT@P [51], (3) Mn/V oxide @MWCNT [56], (4) graphene microspheres [57], (5) laser-induced graphene [58], (6) G-CNT-5 [59], and (7) laser-induced graphene [60].

**Figure 6 molecules-28-01831-f006:**
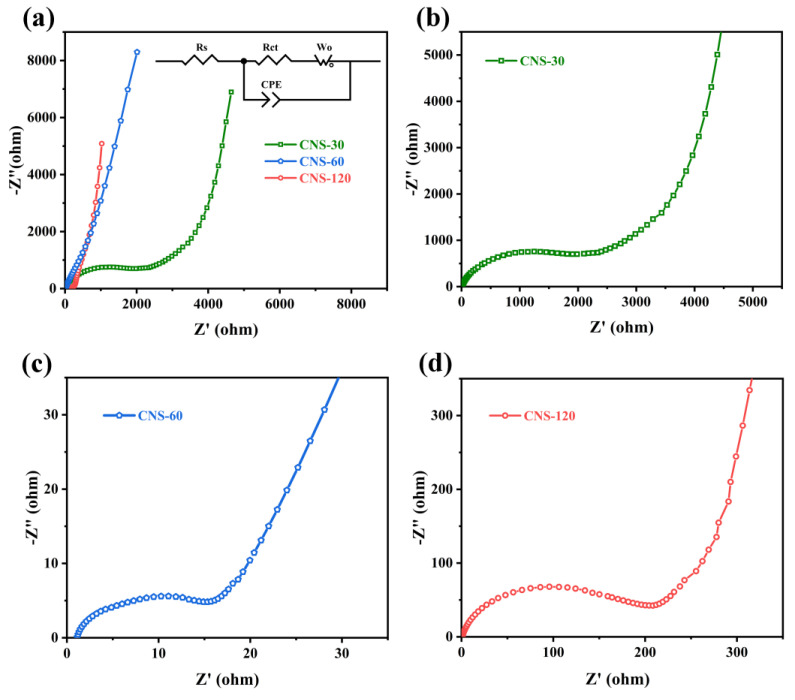
Nyquist plots of all of the samples. (**a**) Nyquist plots of CNS-30, CNS-60, and CNS-120. (**b–d**) The enlarged view of Nyquist plots of CNS-30, CNS-60 and CNS-120 at high frequency, respectively.

**Figure 7 molecules-28-01831-f007:**
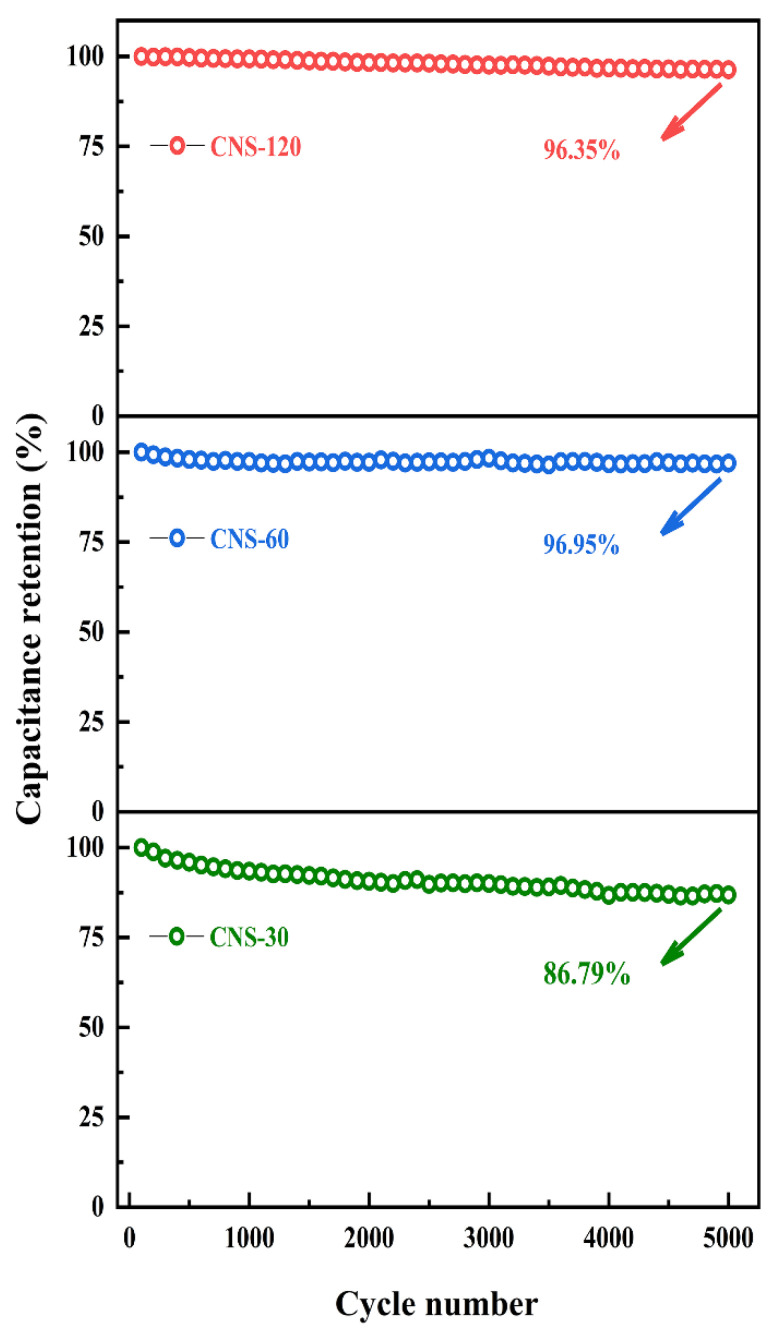
Cycle stability at a current density of 0.1 mA cm^−2^ for 5000 cycles.

**Figure 8 molecules-28-01831-f008:**
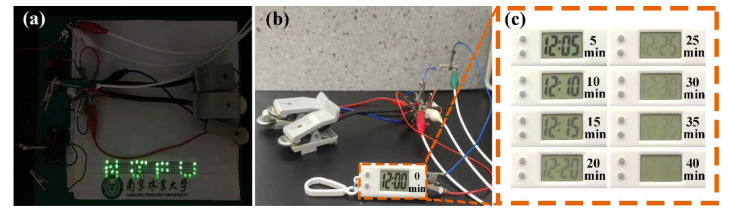
Digital photographs of the application of supercapacitors assembled by CNS-120. (**a**) LED lamps of ‘NJFU’ powered by the coin-type devices. (**b**) An electronic watch powered by the coin-type devices. (**c**) The running time of the watch.

## Data Availability

The data that support the findings of this study are available from the corresponding author upon reasonable request.

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
