# Peer review of "Al Foil-Supported Carbon Nanosheets as Self-Supporting Electrodes for High Areal Capacitance Supercapacitors"

_molecules, 2023, doi:10.3390/molecules28041831_

Round 1
Reviewer 1 Report
The article "Al foil supported carbon nanosheets as self-supporting electrodes for high areal capacitance supercapacitors" presents the synthesis of binder-free electrode materials grown on the Al foil. The subject is interesting, but the manuscript lacks some details and clarifications. So I am suggesting the following points to upgrade the manuscript to meet the journal's standards.
1. The authors synthesized CNS-30, CNS-60, and CNS-120. The authors present an increase in electrochemical performance with concentration. What will be the performance at further higher concentrations?
2. In section 2.1.1, the authors used acetonitrile as a carbon source. The authors should mention the amount of acetonitrile used during the CVD processes.
3. The authors should check equation 3.
4. The Rct value of CNS varies by almost ten times for each sample. The authors need to add a further explanation to support their results.
5. The authors should provide the high-resolution XPS spectrum of zinc and aluminum.
Reviewer 2 Report
It is a good article and it can be published in Molecules.
1- The authors should describe the preparation of the electrode in more detail. Is the system two or three electrodes?? And...
2- What is the active mass?
3- In my opinion, CV and GCD are not compatible. Authors should check the analyzes twice!!
4- What is the applied potential in GCD?(I/g)
5- spacific capacitance is obtained from what formula?
6- Capacitance should be calculated from both CV and GCD analysis. Are these slightly different?
7- Energy storage mechanism should be fully described in one section.
8- Place the equivalent circuit in the Nyquist diagram.
9- Explain more about the carbon in the electrode structure. What is the electrochemical active surface area of this carbon or how much is its electrical conductivity?
10- Compare your research results with other articles in a table.
11- State in the introduction what is the innovative aspect of this research?
